# Effect of Rice, Wheat, and Mung Bean Ingestion on Intestinal Gas Production and Postprandial Gastrointestinal Symptoms in Non-Constipation Irritable Bowel Syndrome Patients

**DOI:** 10.3390/nu11092061

**Published:** 2019-09-03

**Authors:** Sittikorn Linlawan, Tanisa Patcharatrakul, Nicha Somlaw, Sutep Gonlachanvit

**Affiliations:** 1Division of Gastroenterology, Department of Medicine, King Chulalongkorn Memorial Hospital, Thai Red Cross Society, Bangkok 10330, Thailand; 2Center of Excellence on Neurogastroenterology and Motility, Department of Medicine, Faculty of Medicine, Chulalongkorn University, Bangkok 10330, Thailand; 3Division of Clinical Nutrition, Department of Medicine, King Chulalongkorn Memorial Hospital, Thai Red Cross Society, Bangkok 10330, Thailand

**Keywords:** irritable bowel syndrome, rice, wheat, gluten, mung bean, intestinal gas, hydrogen, gastrointestinal symptom

## Abstract

The aim of this study is to evaluate the effect of rice, mung bean, and wheat noodle ingestion on intestinal gas production and postprandial gastrointestinal (GI) symptoms in non-constipation irritable bowel syndrome (IBS) patients. Methods: Twenty patients (13 F, 46 ± 11 y) underwent 8 h breath test studies and GI symptom evaluations after standard rice, wheat, or mung bean noodle meals at 8:00 a.m. in a randomized crossover study with a 1-week washout period. The same meal was ingested at 12:00 p.m. Results: The H_2_ and CH_4_ concentration in the breath samples were similar at baseline (rice:wheat:mung bean, H_2_ = 3.6 ± 0.5:4.1 ± 0.5:4.0 ± 0.7 ppm, CH_4_ = 1.3 ± 0.3:2.1 ± 0.4:1.9 ± 0.4 ppm, *p* > 0.05). Beginning at the fifth hour after breakfast, H_2_ and CH_4_ concentrations significantly increased after wheat compared to rice and mung bean (8 h AUC H_2_ = 4120 ± 2622:2267 ± 1780:2356 ± 1722, AUC CH_4_ = 1617 ± 1127:946 ± 664:943 ± 584 ppm-min, respectively) (*p* < 0.05). Bloating and satiety scores significantly increased after wheat compared to rice (*p* < 0.05), and increased but did not reach statistical significance compared to mung bean (*p* > 0.05). A higher bloating score after wheat compared to rice and mung bean was observed clearly after lunch but not after breakfast. Conclusion: Wheat ingestion produced more intestinal gas and more bloating and satiety scores compared to rice and mung bean, especially after lunch. This provides insight into the role of intestinal gas in the development of bloating symptoms in IBS.

## 1. Introduction

Irritable bowel syndrome (IBS) is a functional gastrointestinal disorder (FGID) that has chronic and disturbing effects on a patient’s life. Complaints of gastrointestinal (GI) symptoms after food ingestion have been reported in 25–64% of IBS patients [1,2,3]. Gas problems and abdominal pain are the most frequent food-related symptoms, and carbohydrates and fats are the most frequent reported causes [1].

Foods can induce GI symptoms by several mechanisms, including exaggerated response to food, food allergy, food intolerances, and increased intestinal gas production [4]. Symptoms induced by an exaggerated response to food and food allergy can develop early after food ingestion, whereas food intolerance and increased intestinal gas production can induce GI symptoms later. The latter effect of food can induce GI symptoms from distention of the gut wall by the liquid and gas volume generated by the osmotic effects of foods or their digested substances and gut fermentation [5], which is the main mechanism of fermentable oligosaccharides, disaccharides, monosaccharides, and polyol(FODMAP)-induced GI symptoms.

The effects of food ingestion on clinical symptoms of IBS patients have mainly been studied in Western countries [6,7,8]. It has been reported that a low FODMAP diet plays a major role in the management of IBS, including a recent study in Asia [9,10,11]. A low FODMAP diet has been reported to reduce IBS symptoms, including bloating, flatulence, abdominal pain, and global symptoms in IBS patients [9,12,13,14]. Most of the low FODMAP studies investigate the effect of low FODMAP diet in IBS patients by looking at a group of foods, rather than individual food items. In selected IBS patients with fructose malabsorption, restriction of fructose, fructans, and polyols for 2–40 months produced a reduction of GI symptoms in 74% of the patients [15]. The effects of an individual source of complex carbohydrates on GI symptoms or intestinal gas production has not been sufficiently investigated, particularly in Asia [16]. In addition, the interplay between the early effect of food ingestion and the latter effect of intestinal fermentation on the development of GI symptoms has not been clearly understood.

Rice and wheat are major sources of carbohydrates. In healthy humans, rice is completely absorbed in the small bowel and produces less intestinal gas after ingestion compared to wheat and other sources of carbohydrates [17]. A previous study demonstrated that a high FODMAP diet produced a different effect on intestinal gas production and GI symptoms when comparing healthy humans and IBS patients [18]. Whether rice is completely absorbed and produces less gastrointestinal gas production and symptoms compared to wheat in IBS has not been explored.

Cellophane noodle is a traditional Asian food. In Thailand and China, it is made from mung bean flour, a complex carbohydrate which is widely available across Asia. The Monash University FODMAP diet application classifies mung bean as a high FODMAP food item (Monash University FODMAP diet^®^, version 3.0.3, Monash University, Melbourne, Australia). The application indicates that mung bean contains a high amount of oligos-GOS and fructans and that intake should be avoided in IBS patients. However, oligosaccharides in mung bean are soluble in water and can be eliminated by adequate presoaking during the process of making the cellophane noodle [19]. However, the effect of processed mung bean or cellophane noodle ingestion on intestinal gas production and GI symptoms in IBS patients has not been fully explored.

In this study, the primary objective was to evaluate the effect of rice, mung bean, and wheat ingestion in the form of noodles on intestinal gas production and postprandial IBS symptoms. We hope to gain insight into the effect of intestinal gas production and postprandial GI symptoms in IBS.

## 2. Materials and Methods

### 2.1. Study Subjects

Non-constipation IBS (non-C IBS) patients, according to the Rome III criteria [20], were recruited from the out-patient clinic of the Center of Excellence on Neurogastroenterology and Motility, Department of Medicine, Chulalongkorn University, Bangkok, Thailand. Exclusion criteria were previous abdominal surgery, pregnancy, active smoking, history of allergy to the test meals, and comorbid conditions such as diabetes, GI malignancy, and pulmonary diseases. Celiac disease, and gluten and wheat allergy were excluded by a serologic test (negative serum immunoglobulin A anti-tissue transglutaminase; tTG antibody) and a specific skin prick test for gluten and wheat (ALK Abello Pharm., Inc., Mississauga, ON, Canada). None of the subjects took probiotic supplements, prokinetics, laxatives, antibiotics, or medications that affect gastrointestinal functions and symptoms during the last 4 weeks prior to the study.

The patients with constipation-predominant IBS defined by Rome III criteria were identified and excluded from this study using a symptom questionnaire with the Bristol Stool Form Scale (BSFS).

### 2.2. Study Design

Participants were randomly assigned by block randomization to ingest three interventional meals (rice, wheat, or mung bean noodle) in a randomized crossover fashion with a one-week washout period. After overnight fasting, GI symptom scores and BSFS were assessed at baseline, then the study meal was given at 8:00 a.m. (breakfast) and 12:00 p.m. (lunch). Although the rice, wheat, and mung bean noodle looked different, all patients were unaware of the main component of the noodles, and the investigators who evaluated symptoms and measured the breath hydrogen (H_2_) and methane (CH_4_) levels were blinded to the test meals. Exhaled breath specimens and GI symptom scores were acquired after the first standard meal at every 15 min for 8 h. The GI symptoms, including bloating, satiety, abdominal pain, abdominal burning, heartburn, urgency of stool, nausea, food regurgitation, acid regurgitation, belching, chest pain/discomfort, and flatulence, were evaluated using 10 cm visual analogue scales (VAS), where 0 indicated no symptoms and 10 represented the worst symptoms. All subjects gave their informed consent for inclusion before they participated in the study. The study was conducted in accordance with the Declaration of Helsinki, and the protocol was approved by the Ethics Committee of Faculty of Medicine, Chulalongkorn University (Project identification code 027/56).

### 2.3. Interventional Meals

Every study meal was made from 90 g dry weight of rice, wheat, or mung bean flour and cooked in noodle form. No vegetables or other poorly absorbed ingredients were added so as to avoid intestinal gas production from other sources. For each study, the same noodle meal made from rice, wheat, or mung bean was given for breakfast and lunch in a randomized cross-over fashion. A glass of water (240 mL) was ingested with the test meals. No food or drink was allowed between meals or after lunch. According to the United States Department of Agriculture (USDA) Food Composition Databases, the total energy in each serving size (90 gm dry weight) for rice, mung bean, and wheat noodle was 327.6, 319.5, and 324.9 kcal, respectively. The amounts of CHO:protein:fat content in a serving size of rice, mung bean, and wheat noodle were 72.2:5.4:0.5 gm, 77.5:0.14:0.05 gm, and 65.3:10.8:1.49 gm, respectively (https://ndb.nal.usda.gov/ndb/search/list).

### 2.4. Breath Tests

Breath samples were collected using a 250 mL sample holding bag (Quintron Instrument Co., Inc., Milwaukee, WI, USA). Subjects were asked to maintain good oral hygiene during the breath testing phase by brushing their teeth before taking their first breath sample and to refrain from vigorous physical activity [21]. On the evening before the study date, all subjects were asked to control the amount of carbohydrate ingestion and avoid poorly absorbable carbohydrates, which may affect the hydrogen breath test study the next morning. The baseline gas sample was a fasting sample taken before breakfast ingestion. Breath samples were analyzed for CO_2_, H_2_, and CH_4_ using a Quintron MicroLyzer Model DP Plus (Quintron Instrument Co., Inc., Milwaukee, WI, USA). The amount of H_2_ and CH_4_ in a breath sample were reported in parts per million (ppm).

### 2.5. Statistical Analysis

The primary endpoint was the difference between the intestinal H_2_ and CH_4_ production after rice, wheat, and mung bean noodle ingestion evaluated from the breath samples collected over an 8 h study period after breakfast. Secondary endpoints were GI symptom score differences after the ingestion of different interventional meals. The sample size was calculated based on the best data available from a previous study of H_2_ production after rice ingestion in healthy volunteers [17]. To determine at least a 30% difference between rice and wheat or mung bean with 90% power at α = 0.05, at least 16 subjects were needed for this cross-over study.

Comparison of H_2_, CH_4_, and GI symptoms involving more than 2 groups were analyzed by the repeat-measures ANOVA and Wilcoxon signed-rank test for parametric and non-parametric data, respectively. A *p*-value of less than 0.05 was defined as statistically significant. Data were expressed as mean ± SD, unless stated otherwise. The data were analyzed using SPSS software version 17.0 for Microsoft Windows.

## 3. Results

Twenty non-C IBS patients (13 F, age 46 ± 11 years) were included. The duration of IBS symptoms before the first diagnosis of IBS was 10.6 ± 0.9 months, and the patients’ BMI was 23.3 ± 4.0 kg/m^2^. All patients completed the studies without any adverse events. All subjects completed the assigned study meals. The patients’ baseline symptom scores are shown in Table 1. Baseline symptom score before each study arm for each GI symptom was not significantly different comparing among the rice, wheat, and mung bean study arm (*p* > 0.05).

The H_2_ and CH_4_ concentrations in breath samples was similar at baseline before breakfast ingestion when comparing the rice, wheat, and mung bean (*p* > 0.05) (Figure 1 and Figure 2). Beginning at 285 min after breakfast, H_2_ and CH_4_ concentration in breath samples for rice noodles were significantly lower than those of wheat noodles (*p* < 0.05), and the difference of H_2_ and CH_4_ concentration persisted until 450 min and 420 min, respectively, after breakfast ingestion. However, H_2_ and CH_4_ concentration in the breath samples for mung bean noodle were significantly lower than those of wheat noodle beginning from 240 min and 270 min until 450 min and 465 min after breakfast ingestion, respectively (*p* < 0.05). The H_2_ and CH_4_ concentrations in breath samples after mung bean and rice ingestions were similar throughout the study period (*p* > 0.05) (Figure 1 and Figure 2).

The mean and area under the curve (AUC) of H_2_ and CH_4_ concentrations over 8 h after wheat noodle ingestion were significantly greater compared to those after rice and mung bean noodle ingestion (*p* < 0.05) (Table 2). The maximum H_2_ levels after wheat ingestion were significantly higher than those after rice and mung bean ingestion (*p* < 0.05). Likewise, the maximum CH_4_ levels after wheat ingestion were significantly higher than those after rice ingestion, but did not reach statistical significance compared to mung bean (Table 2).

The average symptom scores during 8 h after breakfast for bloating and satiety significantly increased after wheat ingestion compared to rice ingestion (wheat vs. rice = 3.0 ± 0.6 vs. 2.2 ± 0.6 and 3.4 ± 0.5 vs. 2.5 ± 0.5, respectively, *p* < 0.05). Although the bloating and satiety symptom scores were higher after wheat ingestion compared to mung bean, the difference did not reach statistical significance (wheat vs. mung bean = 3.0 ± 0.6 vs. 2.3 ± 0.5 and 3.4 ± 0.5 vs. 2.9 ± 0.5, respectively, *p* > 0.05) (Figure 3).

When the bloating symptom severity scores were plotted over time after breakfast ingestion, the bloating symptom scores increased immediately after each meal compared to before the meal, with a greater increase of symptom scores after lunch compared to breakfast, but did not reach statistical significance (*p* > 0.05) (Figure 4). The bloating symptom scores after wheat were significantly higher than rice at 165, 195, 210, 315, 330, 345, 360, and 375 min, and significantly higher than mung bean at 165 and 330 min (*p* < 0.05) after breakfast.

Figure 5 shows the satiety symptom scores over 8 h after breakfast for the rice, mung bean, and wheat study arm. Immediately after meals, the satiety symptom scores increased similarly between breakfast and lunch (*p* > 0.05) and gradually decreased toward the next meal. The improvement of satiety symptom severity scores were similar after breakfast, but after lunch, the improvement was significantly slower for wheat in relation to rice, leading to significantly higher satiety symptom scores after wheat ingestion compared to rice at 330, 345, 360, 375, 390, 420, and 435 min after breakfast, respectively (*p* < 0.05).

Other GI symptoms, including abdominal pain, abdominal burning, nausea, urgency of defecation, heartburn, belching, and regurgitation, were not significantly different between rice, mung bean, and wheat after ingestion (Figure 3).

The H_2_ and CH_4_ production of every 15 min time period (defined as the AUC of each 15 min period) evaluated during the 8 h study periods correlated significantly with the corresponding bloating symptom severity score after the wheat test meal (r = 0.61 and 0.68, respectively, *p* < 0.001), but there was no significant correlation with rice (r = 0.06 and 0.18, respectively, *p* > 0.05) or with mung bean (r = 0.20 and 0.32, respectively, *p* > 0.05). There was no significant correlation between H_2_ and CH_4_ production and satiety symptoms after wheat, rice, and mung bean meals (H_2_, r = −0.15, −0.30, −0.08; CH_4_, r = −0.18, −0.19, 0.1, respectively, *p* > 0.05).

## 4. Discussion

Our study investigated the effects of the ingestion of 3 common carbohydrate sources in noodle form—rice, mung bean (cellophane), and wheat—on intestinal gas production and postprandial gastrointestinal symptoms in non-C IBS patients. We performed simultaneous monitoring of intestinal gas (H_2_ and CH_4_) production and GI symptoms for 8 h after breakfast and lunch. We found that rice and mung bean noodles produced similar levels of H_2_ and CH_4_ in breath samples and a similar severity of postprandial gastrointestinal symptoms, whereas wheat noodles produced higher levels of intestinal gas and a higher bloating symptom severity score compared to rice and mung bean. The satiety symptom scores also significantly increased after wheat noodle ingestion compared to rice noodle ingestion, and increased but did not reach statistical significance compared to mung bean noodle ingestion.

The finding that intestinal gas production significantly correlated with bloating symptom severity scores only after wheat ingestion but not after rice and mung bean suggests that the development of bloating symptoms after wheat was associated with intestinal gas. The finding of no correlation between intestinal gas production and bloating symptom after rice and mung bean agrees with the finding of no significant increase of intestinal gas production compared to baseline after rice and mung bean. No significant correlation between intestinal gas production and satiety symptom severity score after wheat ingestion may be explained by the fact that the immediate effects of food ingestion or stomach distention were the main contribution to satiety symptom development, as shown in Figure 5.

An increase in intestinal gas production measured by H_2_ and CH_4_ was clearly observed, beginning at the end of the fourth hour after breakfast or immediately after lunchtime. This suggests a clearance of incomplete absorbed carbohydrates (CHO) from the ileum into the large bowel stimulated by lunch ingestion. The bloating symptom score increased immediately after breakfast to a similar degree between rice, mung bean, and wheat, and gradually decreased towards the next meal (lunch) (Figure 4). Immediately after lunch, the bloating symptom score increased compared to before lunch for every test meal and gradually decreased. However, the increase of bloating symptom scores after wheat ingestion was maintained longer than rice and mung bean, and led to significantly higher bloating symptom scores when compared to rice or mung bean during the 90–120 min after lunch. This finding indicates the effects of an increase in intestinal gas production on bloating symptoms after a meal. According to the results of this study, we proposed a model of bloating symptom generation by food ingestion and intestinal gas production as a function of 3 main factors: (1) direct meal volume effect, which affects immediately after each meal ingestion; (2) effect of intestinal gas from new CHO arriving at the colon secondary to ileal emptying, which is available and contributes to symptoms after the lunch; and (3) effect of intestinal gas from residue CHO in the colon from the previous meal, which is available before dinner. This model suggests bloating symptoms should be less apparent in the morning (after overnight absorption of the intestinal gas into the circulation) and get worse after lunch and dinner if the patient consumes poorly absorbed CHO or high FODMAP for every meal of the day. It has been reported that bloating and abdominal distention in IBS patients was lower in the morning but significantly higher at the end of the day [22]. Abdominal girth monitoring studies demonstrated that 46% of IBS patients exhibited maximum abdominal girth at the end of the day and 24% at the time of meal ingestion [22]. Our finding suggests that bloating symptoms and abdominal distention, which are low in the morning, increase in the afternoon, and get worse in the evening or after dinner, may be characteristics of IBS patients who are likely to suffer from an increase in intestinal gas production from food ingestion.

The finding that, after lunch, the improvement of satiety symptom after wheat was delayed compared to rice suggests that an increase in intestinal gas may prolong postprandial satiety symptoms.

The Monash University FODMAP diet application classified rice as a low FODMAP food. In contrast, mung bean and wheat are classified as high FODMAP foods. Rice and mung bean are major sources of carbohydrates in Asia, whereas wheat is a major source of carbohydrates in Western countries [16]. We found that wheat noodle produced higher intestinal gas, both H_2_ and CH_4_, than rice and mung bean noodles. The low intestinal gas production after mung bean noodle ingestion found in this study indicates the elimination of oligosaccharides by adequate presoaking during the process of making mung bean noodle, whereas the wheat noodle preparation process did not provide adequate elimination of the FODMAP contents.

The different effects on intestinal gas production and gastrointestinal symptoms of rice and mung bean compared to wheat could be seen at the beginning of the fifth hour after breakfast ingestion. This result agrees with other studies that demonstrated the positive effects of a low FODMAP diet on IBS symptoms that appear early, within 1 or 2 days [18].

Rice, wheat, and mung bean are 3 major sources of carbohydrates. Rice and wheat rank high on the glycemic index [23,24], whereas mung bean ranks low on the glycemic index. Ingestion of mung bean has been reported to improve diabetes control and protein reservation in type 2 diabetes mellitus [25]. This study supports mung bean as a better source of CHO for IBS patients with diabetes mellitus compared to rice and wheat. It has been reported that 10% of IBS patients have diabetes [26].

Previous studies showed that GI symptoms, especially bloating and early satiety, after wheat ingestion resulted not only from incomplete intestinal absorption but also from osmotic load and luminal distention in the small or large intestine, which can trigger symptoms by increasing small bowel water content, confirmed by abdominal MRI [5].

The limitations of our study are: (1) we could not make the 3 different study meals look similar; (2) ingestion of meal in the evening before the study date may affect intestinal gas production after breakfast in this study; (3) the sample size of 20 was relatively small to demonstrate the difference in severity of some symptoms; and (4) the amount of CHO:protein:fat per gram dry weight in rice, mung bean, and wheat noodle was not identical. However, we tried to minimize these limitations by not informing the patients of the CHO sources and expected effects, using a crossover method with a 1-week washout period, by blinding the study meals to the investigators, and by asking the patients to record their diet on the day before the study. The fact that there is less CHO is in wheat noodle than rice and mung bean noodle, which supports our conclusion that wheat CHO produces more gas than rice and mung bean, is not a limitation.

We excluded patients with constipation because it has been reported that constipation patients may harbor preformed gas in hard stools, and the gas can be released when mixing of the intestinal content is induced [27]. In our study, no patients had a BSFS of 1 or 2, which has been reported to be associated with preformed gas in the stool [27].

In conclusion, this study suggests that rice and mung bean are better sources of carbohydrates for non-C IBS than wheat carbohydrate, and they produce less intestinal gas and fewer bloating and satiety symptoms. Increased intestinal gas production after wheat, a high FODMAP food item, developed obviously after the second meal (lunch ingestion) and was associated with more postprandial bloating and satiety symptoms after lunch. This study provides insight into the role of intestinal gas production on the development of postprandial bloating and satiety symptoms in non-C IBS patients.

## Figures and Tables

**Figure 1 nutrients-11-02061-f001:**
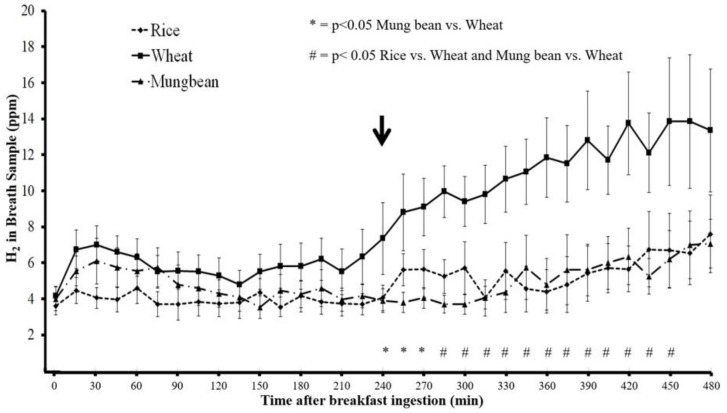
Intestinal hydrogen (H_2_) gas levels (mean ± SEM, ppm) measured from breath samples in non-constipation irritable bowel syndrome (non-C IBS) patients after ingestion of rice, wheat, and mung bean noodle (arrow = time of lunch ingestion).

**Figure 2 nutrients-11-02061-f002:**
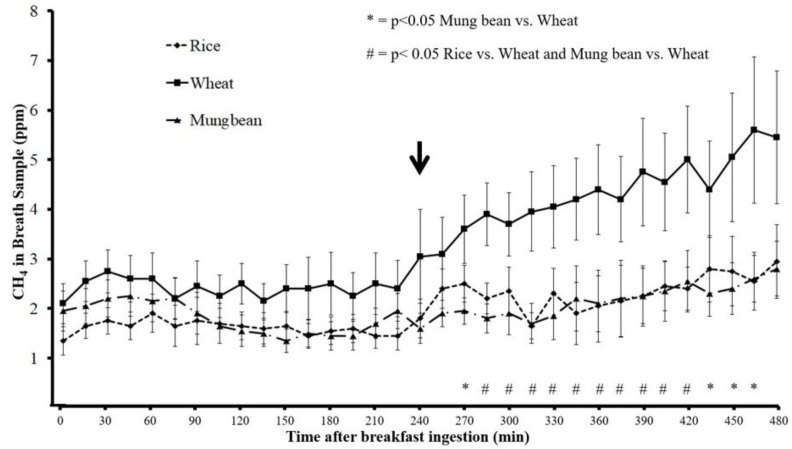
Intestinal methane (CH_4_) gas levels (mean ± SEM, ppm) measured from breath samples in non-C IBS patients after ingestion of rice, wheat, and mung bean noodle (arrow = time of lunch ingestion).

**Figure 3 nutrients-11-02061-f003:**
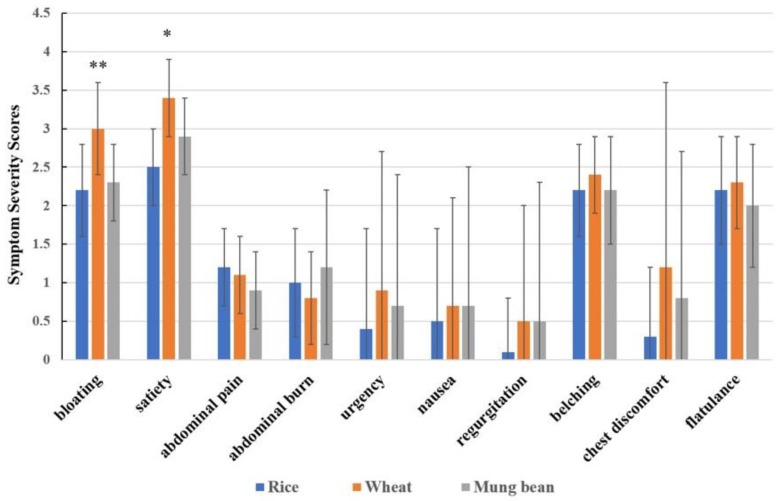
Gastrointestinal symptom scores (visual analogue scale, VAS 0–10) in non-C IBS patients after ingestion of different test meals (data expressed as mean ± SEM). ** = *p* < 0.05 wheat vs. rice and wheat vs. mung bean, * = *p* < 0.05 wheat vs. rice.

**Figure 4 nutrients-11-02061-f004:**
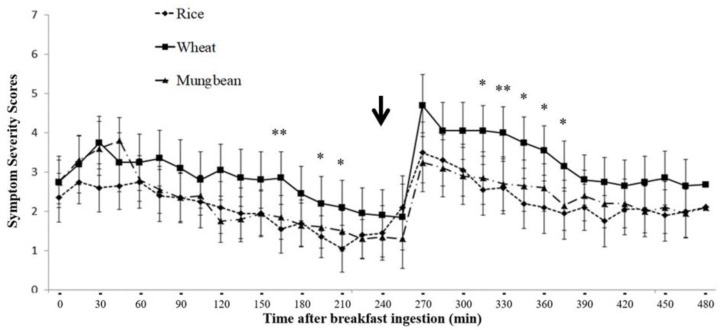
Abdominal bloating symptom severity score at baseline and every 15 min after breakfast until the end of the breath test study (data expressed as mean ± SEM) (arrow = time of lunch ingestion); * = *p* < 0.05 wheat vs. rice, ** = *p* < 0.05 wheat vs. rice and wheat vs. mung bean.

**Figure 5 nutrients-11-02061-f005:**
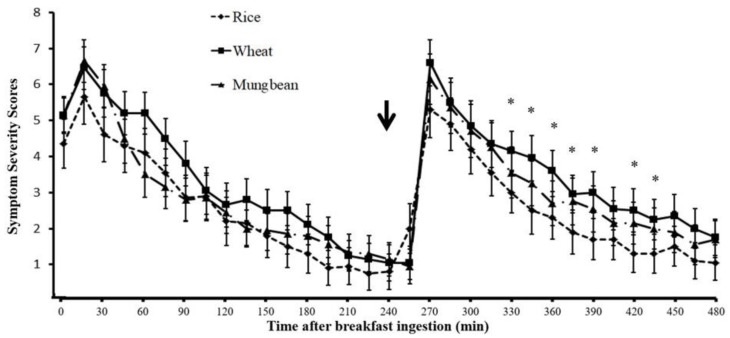
Satiety symptom severity score at baseline and every 15 min after breakfast until the end of the breath test study (data expressed as mean ± SEM) (arrow = time of lunch ingestion); * = *p* < 0.05 wheat vs. rice.

**Table 1 nutrients-11-02061-t001:** Baseline gastrointestinal symptom scores before breakfast for rice, wheat, and mung bean study arm (data expressed as mean ± SEM) *.

Symptoms	Rice	Wheat	Mung Bean
Bloating	2.3 ± 0.5	2.7 ± 0.6	2.8 ± 0.5
Satiety	4.3 ± 0.7	5.1 ± 0.5	5.1 ± 0.5
Abdominal pain	1.3 ± 0.5	1.0 ± 0.5	1.5 ± 0.6
Abdominal burning	1.1 ± 0.4	0.7 ± 0.3	1.8 ± 0.5
Urgency to defecate	0.7 ± 0.4	0.6 ± 0.4	0.7 ± 0.3
Heartburn	0.5 ± 0.3	0.0 ± 0.0	0.3 ± 0.2
Nausea	0.8 ± 0.4	0.6 ± 0.4	1.4 ± 0.6
Regurgitation	0.3 ± 0.3	0.6 ± 0.4	0.9 ± 0.5
Belching	1.9 ± 0.6	2.1 ± 0.7	2.1 ± 0.7
Chest discomfort	0.2 ± 0.2	0.5 ± 0.5	0.6 ± 0.4
Flatulence	1.9 ± 0.7	1.8 ± 0.7	2.4 ± 0.7

* = Not significantly different (*p* > 0.05).

**Table 2 nutrients-11-02061-t002:** Intestinal gas production after ingestion of study diet (rice, wheat, and mung bean noodle).

Intestinal Gas	Rice	Wheat	Mung Bean
**Mean Hydrogen (ppm)**	4.8 ± 3.8	8.7 ± 5.6 *	4.9 ± 3.6
**Mean Methane (ppm)**	2.0 ± 1.4	3.4 ± 2.4 *	2.0 ± 1.2
**Peak Hydrogen (ppm)**	11.7 ± 9.3	21.5 ± 16.3 *	11.9 ± 10.1
**Peak Methane (ppm)**	4.8 ± 3.4	8.2 ± 6.3 ^#^	5.4 ± 3.2
**AUC Hydrogen (ppm-min)**	2267 ± 1780	4120 ± 2622 *	2356 ± 1722
**AUC Methane (ppm-min)**	946 ± 664	1617 ± 1127 *	943 ± 584

* = *p* < 0.05 wheat vs. rice and mung bean, ^#^ = *p* < 0.05 wheat vs. rice. AUC; area under the curve

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
