# Peer review of "Effect of Rice, Wheat, and Mung Bean Ingestion on Intestinal Gas Production and Postprandial Gastrointestinal Symptoms in Non-Constipation Irritable Bowel Syndrome Patients"

_nutrients, 2019, doi:10.3390/nu11092061_

Round 1
Reviewer 1 Report
Interesting paper. I have some concerns regardind the sample size: how did the authors determine it? Please specify.
Then, I suggest to read and cite a very recent paper regarding the "programming" of FGID:
Neonatal Antibiotics and Prematurity Are Associated with an Increased Risk of Functional Gastrointestinal Disorders in the First Year of Life.
Salvatore S, Baldassarre ME, Di Mauro A, Laforgia N, Tafuri S, Bianchi FP, Dattoli E, Morando L, Pensabene L, Meneghin F, Dilillo D, Mancini V, Talarico V, Tandoi F, Zuccotti G, Agosti M.
J Pediatr. 2019 Jun 11. pii: S0022-3476(19)30544-X. doi: 10.1016/j.jpeds.2019.04.061.
Author Response
The sample size was calculated based on the best data available from a previous study of H2 production after rice ingestion in healthy volunteers (Levitt MD, Hirsh P, Fetzer CA, Sheahan M, Levine AS. H2 excretion after ingestion of complex carbohydrates. Gastroenterology 1987; 92:383-9.) to determine at least a 30% difference between rice and wheat or mung bean with 90% power at α = 0.05. Thanks for the suggestion of the paper.
Reviewer 2 Report
This is an interesting cross-over study on the effects of rice, wheat and mung bean meals on intestinal gas production and GI symptoms in 20 non-constipated IBS patients.
Comments and suggestions: 1) The title is misleading: patients ingested not only carbohydrates but noodles 2) Please explain „exaggerated response to food“3) Please specify allergy testing. What antigens were used in the skin prick test and did it reallly exclude gluten allergy? 4) What else were the patients allowed to eat or drink during the study period apart from the noodle? 5) Statistics: were there any correlations between symptoms and gas levels?
6) Maybe the authors can comment on the relatively short duration of IBS symptoms in their patients and the overall rather low symptom scores reported. It would be helpful if a standardized score like IBS-SSS was available. 7) I have some difficulty to understand the main symptom „satiety“ which in contrast to e.g. fullness is not a bad thing in itself, especially after a meal. I know "early satiety“ as a symptom of IBS but this would be be in conflict with the high baseline scores after overnight fasting. Please clarify. 8) The discussion could be substantially shortened by eliminating the repetition of reporting results.
Author Response
1) The title is misleading: patients ingested not only carbohydrates but noodles
Response1: Thanks for your comment. Although the study aim is to compare the effect of different carbohydrate which is the main component of the interventional meals on intestinal gas production, the test meals contained small amount of protein and fat. So, we agree to remove “carbohydrates” from the title and changed to “Effect of Rice, Wheat and Mung Bean Ingestion on Intestinal Gas Production and Postprandial Gastrointestinal Symptoms in Non-constipation Irritable Bowel Syndrome Patients.
2) Please explain “exaggerated response to food”
Response 2: “exaggerated response to food” is the abnormal or over-reaction to food that leads to GI symptoms. It can be explained by many mechanisms such as visceral hypersensitivity, abnormal gut-brain interaction, and abnormal GI motility reflex responses such as gastrocolonic reflex.
3) Please specify allergy testing. What antigens were used in the skin prick test and did it really exclude gluten allergy?
Response 3: We performed skin prick test for wheat and gluten to exclude wheat and gluten allergy. Celiac disease was excluded by a serologic test (negative serum immunoglobulin A anti-tissue transglutaminase; tTG antibody). We added this information in the revised manuscript.
4) What else were the patients allowed to eat or drink during the study period apart from the noodle?
Response 4: The patients could drink a glass of water during the meal and did not allow to take any food or drink between meals. This information was added in the revised manuscript.
5) Statistics: were there any correlations between symptoms and gas levels?
Response 5: Thank you for this excellent suggestion. We found that the bloating symptom significantly correlated with H2 and CH4 concentrations for wheat but not for rice and mung bean. This was added in the results and discussion in the revised manuscript.
6) Maybe the authors can comment on the relatively short duration of IBS symptoms in their patients and the overall rather low symptom scores reported. It would be helpful if a standardized score like IBS-SSS was available.
Response 6: Thanks for your valuable comment. The symptom severity scores were rather low because the baseline symptom scores were evaluated in the morning before breakfast. This agrees with the nature of symptoms in IBS with bloating which is usually mild after wake up in the morning and getting worse in the afternoon and evening.
The duration of IBS symptoms in this study was 11 months which is the duration before the first diagnosis of IBS. This duration was adequate for IBS diagnosis by Rome criteria. The total duration of IBS in this study was longer as patients were treated for periods of time before including into this study.
We did not use IBS-SSS because we would like to explore both upper and lower GI symptoms to demonstrate the immediate effect of the test meals. The duration of IBS was clarified in the revised manuscript.
7) I have some difficulty to understand the main symptom “satiety“ which in contrast to e.g. fullness is not a bad thing in itself, especially after a meal. I know "early satiety” as a symptom of IBS but this would be in conflict with the high baseline scores after overnight fasting. Please clarify.
Response 7: “Satiety” is the state of being fed to or beyond the capacity and cannot have anymore. Patients may report satiety during fasting period if the symptom is similar to the sensation that occurs after a meal. In this study, we would like to monitor and compare the symptoms over 8 hours and compare between different meals. Thus the word “satiety” is more suitable than “early satiety”. We did not evaluate whether the patients had “early satiety” or not as the size of the test meals might be different from their usual meals.
8) The discussion could be substantially shortened by eliminating the repetition of reporting results.
Response 8: Thank you very much. We revised as your suggestion.
Reviewer 3 Report
I've read with attention the paper by Sittikorn et al. that is potentially of interest. The methodology applies is overall correct, the results are reliable and adequately discussed. I've only a couple of minor comments:
The abstract should include some quantitative data Table 1 is not useful since a half of the data included is already reported in the text. It will be sufficient to report the other 2 data in the text. The discussion is a bit long and unfocused. Moreover, the inclusion of table 4 in the discussion section is a bit strange, since it seems to be more result than part of the discussion.Author Response
Response to reviewer 3 comments
The abstract should include some quantitative data. Table 1 is not useful since a half of the data included is already reported in the text. It will be sufficient to report the other 2 data in the text. The discussion is a bit long and unfocused. Moreover, the inclusion of table 4 in the discussion section is a bit strange, since it seems to be more result than part of the discussion.
Response 1: Thank you for this valuable suggestion. We added the quantitative data in the revised abstract as suggested. The data in table 1 were moved to the text and table 1 was deleted in the revised manuscript. Table 4 in the discussion was removed and the content in table 4 was moved to text in discussion.